# Uptake and effectiveness of a tailor-made online lifestyle programme targeting modifiable risk factors for dementia among middle-aged descendants of people with recently diagnosed dementia: study protocol of a cluster randomised controlled trial (Demin study)

Joyce Vrijsen [1], Ameen Abu-Hanna,[2] Els LM Maeckelberghe,[3] Peter Paul De Deyn,[4] Andrea F de Winter,[5] Fransje E Reesink,[4] Richard C Oude Voshaar,[6] Erik Buskens,[1] Sophia E de Rooij,[7] Nynke Smidt,[1] On behalf of the Demin consortium

For numbered affiliations see end of article.

**Correspondence to**
Joyce Vrijsen; j.vrijsen@umcg.nl

## ABSTRACT

**Introduction** Descendants of patients with dementia have a higher risk to develop dementia. This study aims to investigate the uptake and effectiveness of an online tailor-made lifestyle programme for dementia risk reduction (DRR) among middle-aged descendants of people with recently diagnosed late-onset dementia.

**Methods and analysis** Demin is a cluster randomised controlled trial, aiming to include 21 memory clinics of which 13 will be randomly allocated to the passive (poster and flyer in a waiting room) and 8 to the active recruitment strategy (additional personal invitation by members of the team of the memory clinic). We aim to recruit 378 participants (40–60 years) with a parent who is recently diagnosed with Alzheimer's disease or vascular dementia at one of the participating memory clinics. All participants receive a dementia risk assessment (online questionnaire, physical examination and blood sample) and subsequently an online tailor-made lifestyle advice regarding protective (Mediterranean diet, low/moderate alcohol consumption and high cognitive activity) and risk factors (physical inactivity, smoking, loneliness, cardiovascular diseases (CVD), hypertension, high cholesterol, diabetes, obesity, renal dysfunction and depression) for dementia. The primary outcome is the difference in uptake between the two recruitment strategies. Secondary outcomes are change(s) in (1) the Lifestyle for Brain Health score, (2) individual health behaviours, (3) health beliefs and attitudes towards DRR and (4) compliance to the tailor-made lifestyle advice. Outcomes will be measured at 3, 6, 9 and 12 months after baseline. The effectiveness of this online tailor-made lifestyle programme will be evaluated by comparing Demin participants to a matched control group (lifelines cohort).

### Strengths and limitations of this study

► This is the first multicentre trial that focuses on dementia risk reduction in middle-aged descendants of recently diagnosed patients with Alzheimer's disease or vascular dementia.

► The programme gives participants insight in their risk and protective factors for dementia and provides a tailor-made online lifestyle advice with regard to 13 modifiable risk factors for dementia, taking the stages of (health behaviour) change into account.

► The application ensures the privacy of the participants by using SMS-tan for logging in their personal account and signing the electronic informed consent form.

► The web-based application (demin.nl) functions fully automatically, making it easy to implement the study in other memory clinics and settings.

► Changing health behaviour is difficult and it is unclear whether a tailor-made online lifestyle advice is sufficient to change health behaviour and to maintain a healthy lifestyle.

**Ethics and dissemination** This study has been approved by the Dutch Ministry of Health, Welfare and Sport according to the Population Screening Act. All participants have to give online informed consent using SMS-tan (transaction authentication number delivered via text message). Findings will be disseminated through peer-reviewed journals and (inter)national conferences.
**Trial registration number** NTR7434.

## INTRODUCTION

Dementia is considered as a major public health concern.[1] Due to the ageing population, the number of dementia cases will increase substantially in the next decades. In 2015, more than 46 million people worldwide were affected by dementia and this number is expected to increase to 131 million by 2050.[2] This rise in people with dementia carries a high economic and social burden for society.[1] In 2015, global costs of dementia reached 818 billion US dollars and will increase further.[3] Currently, no curative treatments are available. Therefore, prevention is a key element to counteract the dementia epidemic.[4 5]

The most common types of dementia are Alzheimer's disease (AD) (60%–70%), vascular dementia (VD) (15%–20%) or a combination of AD and VD (mixed dementia).[6–8] The presence of a first-degree relative with AD doubles the risk for developing AD.[9] This increased risk has several reasons. First, descendants of people with AD more often have a higher genetic predisposition for AD (eg, carrier of the apolipoprotein E (APOE) ε4 allele).[9] Second, high blood pressure, vascular diseases and other vascular risk factors (ie, diabetes type 2, obesity, hypercholesterolemia) often cluster in families.[10] Lastly, psychosocial behaviour runs in the family and also affects health behaviour and lifestyle.[11 12] Not surprisingly, individuals with a parent who is recently diagnosed with AD or VD often worry about their own risk of developing dementia. Therefore, this life event (parental diagnosis of dementia) might encourage the willingness of individuals to change their health behaviour.[13]

Parental family history has been associated with an increased risk of dementia independently of known genetic risk factors.[9 14] Therefore, a healthy lifestyle might be beneficial for individuals with a positive family history, especially for APOE ε4 carriers.[15–18] Over the last decade, evidence of modifiable risk factors for dementia has been mounting.[4 6 19] The Lancet commission on dementia prevention, intervention and care demonstrated that 35% of the dementia cases is attributable to modifiable risk factors (ie, less education, hearing loss, midlife hypertension, midlife obesity, smoking, depression, physical inactivity, social isolation and diabetes) and recommended to start interventions including more childhood education, promotion of physical exercise, reduction of smoking, maintaining social engagement and management of hypertension, diabetes, obesity, depression and hearing loss.[4 6 20] Other major risk factors are hyperlipidaemia, coronary heart disease, renal dysfunction, Mediterranean diet and cognitive activity.[19]

Only a few studies examined the effectiveness of targeting these modifiable factors on cognitive decline and dementia incidence through a multidomain intervention, such as the Finnish Geriatric Intervention Study to Prevent Cognitive Impairment and Disability (FINGER) study,[21] the Prevention of Dementia by Intensive Vascular care (PreDIVA) study[22] and the The Multi-domain Alzheimer Preventive Trial (MAPT) study.[23] These studies, with a follow-up varying from 2 to 6 years, found small or

non-significant effects on cognition in older participants (eg, >60 years).[21–23] Starting multidomain interventions earlier in life might be promising as cognitive decline begins already in midlife.[24 25] However, since dementia is mainly prevalent in the elderly, a long follow-up period of approximately 20 years would be required in order to determine the effectiveness of interventions on dementia incidence.[24–26] Furthermore, tailoring interventions improves the effectiveness of health behaviour change interventions.[27] Web-based interventions have the potential to support health behaviour change as there is the opportunity to tailor lifestyle advice.[28–31] They were especially effective when a theoretical basis or conceptual framework (eg, health belief model (HBM), transtheoretical model, theory of planned behaviour, I(integrated)-change model[32–36]), behaviour change techniques (eg, providing feedback on performance and information on the consequences of unhealthy behaviour) and several modes of delivery had been used.[27]

The first challenge of health behaviour change interventions is to achieve a high level of uptake for screening (eg, assessing risk and protective factors for dementia), reflecting the willingness to participate. A systematic review identified a large variation in uptake in health checks and lifestyle intervention programmes,[37] depending on the type of recruitment strategy. The two main types of strategies for recruitment are the active and passive recruitment strategy. Active recruitment involves a personal invitation by the project staff and healthcare providers (eg, proactive) and passive recruitment involves recruitment of participants through various channels such as flyers and advertisements (eg, reactive).[38] The most effective recruitment strategy is a proactive referral from a healthcare provider, while displaying posters and flyers showed to be less effective.[39 40] Uptake also depends on other factors as described in social cognition models (eg, knowledge, perceived susceptibility and severity, facilitators, benefits and barriers and attitude towards such interventions).[32–36] These factors are essential and useful to make a well-informed decision about dementia risk assessment, considering the possible benefits and harms. Therefore, information on dementia, the risk and protective factors for dementia, heritability and how to tackle risk and protective factors for dementia are important factors in the development of a web-based intervention. A previous study showed that the majority of the Dutch general population is unaware of the relationship between modifiable risk factors and brain health, particularly regarding major cardiovascular risk factors (eg, hypertension, hypercholesterolemia and coronary heart disease).[41] It is shown that this lack of knowledge is a barrier to the uptake and maintenance of healthy behaviours for middle-aged individuals.[42] Having a parent who is recently diagnosed with AD or VD could have led to an increased knowledge on dementia and risk perception.[13] Therefore, middle-aged descendants of recently diagnosed people with AD or VD might be receptive to assess their risk and motivated to adopt a healthier lifestyle

as they just realised their (familial) risk.[13 43] Although we expect that the uptake in the active recruitment strategy will be higher compared with the passive recruitment strategy, participants recruited via the passive recruitment strategy might be more intrinsically motivated to adopt and maintain their healthy lifestyle and less likely to drop out of the study.

To our knowledge, none of the health behaviour intervention studies were aimed at a specific group of middle-aged adults with increased risk for dementia due to their parental family history of dementia. Therefore, this study aims to investigate the uptake and effectiveness of a tailor-made online lifestyle programme for dementia risk reduction (DRR) among middle-aged descendants of recently diagnosed (in the last 6 months) people with AD or VD in the Netherlands. This will give insight into what extent it is feasible to recruit middle-aged descendants of people with AD or VD at the memory clinics and whether these potential participants are willing to participate in a tailor-made online lifestyle programme in order to reduce their dementia risk.

## METHODS AND ANALYSIS
### Study setting and design
This study is a pragmatic cluster randomised controlled trial (RCT), including 21 participating memory clinics in the Netherlands who are randomly allocated to a passive or active recruitment of participants. Memory clinics allocated to the active recruitment strategy invite potential participants face-to-face by a member of the team of the memory clinic to participate in the tailor-made online lifestyle programme for DRR (also called the Demin study), next to posters and flyers that are placed in the waiting room of the memory clinic. Memory clinics allocated to the passive recruitment strategy, do not invite potential participants proactively, but invite potential participants to participate in the Demin study by posters and flyers that are placed in the waiting room of the memory clinic.

Patients with AD or VD (or their caregivers) receive an envelope either at the registration desk of the memory clinic or after the consult of the patient (only with active recruitment). This envelope is addressed to the middle-aged descendants of patients with recently diagnosed AD or VD and includes a patient information form (PIF) with information about the content of the study, the advantages and disadvantages of study participation and how potential participants can participate. Potential participants (one family member per patient) are asked to register themselves (eg, making an account) on the Demin website (www.demin.nl), by using the memory clinic specific login access code, which is reported on the front page of the PIF and represents the memory clinic in which the parent was diagnosed. The decision to participate is confirmed by the participants by signing the online informed consent form (electronic signature by using SMS-tan). After signing this form, individuals from both recruitment strategies are able to log in to their personalised website 'My Demin' and continue the intervention in an equal manner. The personalised website 'My Demin' is secured and only accessible for the participant by logging in with their personal email address, password and SMS-tan code. 'My Demin' contains the following information: (1) My personal (account) information, (2) Message inbox, (3) My online questionnaires, and (4) My personal health profile including online tailor-made lifestyle advice. After participants have completed the online questionnaire, they automatically receive a message with a request to make an appointment for physical examination including a fasting blood sample. Moreover, participants can invite siblings to participate in the study in 'My Demin'.

The functionalities provided by the Demin website are based on the literature and input we received from people with a parent with dementia (focus group discussions).

### Randomisation of memory clinics
To prevent contamination between the two recruitment strategies, randomisation is performed at the level of the memory clinics. To enhance comparability between the intervention (participants of the active recruitment strategy) and control group (participants of the passive recruitment strategy), the memory clinics will be matched and randomised by a statistician, who is blind to the identity of the memory clinics and not involved in the study. First, all participating memory clinics will be matched into pairs based on the following criteria: (1) number of newly diagnosed dementia (VD, AD or mixed dementia) patients seen per year (range vary from 60 to 350 patients per year) and (2) the average social economic position (SEP) of the population living around the memory clinic (neighbourhood SEP), based on data from Statistics Netherlands.[44] Second, the matched memory clinics will be randomised (pairwise randomisation) to an active recruitment strategy or passive recruitment strategy using a computer-generated random number list. As we expect a higher response rate in the active recruitment strategy group, we use an active:passive recruitment strategy ratio of 8:13 (see sample size calculations).

### Study population
Eligible participants are middle-aged individuals (40–60 years old) with a parent who is recently (less than 6 months ago) diagnosed with AD or VD (or mixed dementia) at one of the participating memory clinics in the Netherlands (see Acknowledgements section). Individuals should provide informed consent, be able to fill out an online Dutch questionnaire. Pregnant women are excluded from participation.

### Sample size calculations
The primary outcome measure is uptake, which is defined as the percentage of eligible individuals that signed the online informed consent form and completed baseline assessment (online questionnaire and physical examination and a fasting blood sample). In order to detect

a difference of 20% in uptake between the passive and active recruitment strategy (30% vs 50%), we need 94 participants in each group to achieve a power of 80% with alpha levels of 0.05 (total=188 participants). To take cluster randomisation into account, we use the formula $1+((n-1)*ICC)$ (inflation factor), where n is the average number of included participants per memory clinic and ICC is Intra Class Correlation.[45] The ICC is unknown, but an ICC of 0.05 is a common value for cluster RCTs in hospitals.[46] The estimated average of included participants per memory clinic per year is n=15 using a passive recruitment strategy and n=25 using an active recruitment strategy, taking into account non-response. With unequal cluster sizes, 'n' is replaced by 'm', where m is the sum of $(M)^2/sum(M)$ $((15^2+25^2)/(15+25))$.[47] This results in a sample size inflation factor of $(1+((21.25-1)*0.05)=2.01$. Therefore, the total number of participants needed is 378 (2.01*188). In order to recruit 378 participants, we need 21 memory clinics, of which eight memory clinics (responsible for 189 included participants) will be allocated to the active recruitment strategy and 13 memory clinics (responsible for 189 included participants) will be allocated to the passive recruitment strategy.

## Demin website

The Demin website is available for everyone and provides information about dementia, heredity of dementia, risk and protective factors for dementia, and how to tackle potential risk factors for dementia. The health information will be provided by written text and in an audiovisual format, such as a spoken animation, to assure inclusion of participants with different levels of health literacy.[48] According to the cognitive theory of multimedia learning, people process visual and auditory information through different channels.[49 50] It is known that health information provided by various channels, such as written text and spoken animations, improves information processing compared with information only provided through written text or spoken animations.[49 50] The instructions for registration (making an account, signing informed consent) are also provided as written text and visual screenshots representing the steps of the registration process.

## Online tailor-made lifestyle programme for dementia risk reduction

After participants give online informed consent, participants have access to the online tailor-made lifestyle programme for DRR, which consists of (1) a dementia risk assessment and (2) an online tailor-made lifestyle advice including a personal health profile targeting risk and protective factors for dementia.

## Dementia risk assessment

The dementia risk assessment consists of filling out an online questionnaire (in 'My Demin') and physical examination, including a fasting blood sample, at one of the 21 participating memory clinics in order to determine whether risk and protective factors are present. In order

to minimise the amount of missing data, validation and skip-and-fail rules were implemented in the online questionnaire. Furthermore, automatic reminders are sent to the participant if the online questionnaire was not filled in within 2 weeks. Physical examination will be conducted by the team of the local memory clinic and includes the following measurements: height (in cm) (SECA 222 stadiometer), body weight (in kg) without shoes (SECA 761 scale), waist and hip circumference (in cm) (SECA 200 measuring tape) and three measurements of diastolic and systolic blood pressure (in mm Hg) (Welch Allyn 'Spot Vital Signs'[51]). After physical examination, which takes approximately 15 min, a fasting blood sample (maximum of 21 mL) is taken for direct laboratory measurement of glucose, HbA1C, total cholesterol, high-density lipoprotein, low-density lipoprotein, triglycerides and serum creatinine. The results of the physical examination (height, body weight, blood pressure and waist and hip circumference) are sent to the researcher (JV) to check the entry of the results by the participants. The results of the direct laboratory measurements are sent to the medical doctor (EMA) of the University Medical Centre Groningen to check for deviating values.

## Risk and protective factors for dementia

Through the online questionnaire and physical examination, data on 13 currently known protective (ie, Mediterranean diet, low/moderate alcohol consumption, cognitive activity) and risk factors (ie, physical inactivity, smoking, loneliness, cardiovascular diseases, hypertension, high cholesterol, diabetes mellitus, obesity, renal dysfunction, depression) for dementia are collected.[6 19 52] See table 1 for an overview of the assessment measures. The measurements of these risk and protective factors are described in online supplemental file 1.

## Personal health profile

After completion of the baseline dementia risk assessment (including the data entry of the physical examination and laboratory measurements), a personal health profile is automatically provided in the personal account of the participants (My Demin). The personal health profile gives an overview of the presence of the risk and protective factors for dementia, without including the weight of the risk and protective factors. According to the Lifestyle for Brain Health (LIBRA) score, each risk and protective factor[19 52 53] is categorised into one of the following categories: (1) room for improvement, (2) remember to manage well, (3) keep this up (see table 2). The 'keep this up' category represents factors that participants are currently managing well or diseases they do not have. The 'room for improvement' category represents the factors that could be improved by health behaviour change (eg, quit smoking, become more physical active, change diet, drink less alcohol). The category 'remember to manage well' is assigned when a risk factor (ie, cardiovascular diseases, hypertension, high cholesterol, diabetes mellitus, renal dysfunction and depression) is

**Table 1** Assessment measures at baseline and follow-up

| | Baseline | 3 months | 6 months | 9 months | 12 months |
|---|---|---|---|---|---|
| **Risk and protective factors** | | | | | |
| Smoking | Q | Q | Q | Q | Q |
| Physical inactivity (SQUASH, IPAQ) | Q | Q | Q | Q | Q |
| Mediterranean diet (FFQ) | Q | Q | Q | Q | Q |
| Alcohol consumption (FFQ) | Q | Q | Q | Q | Q |
| High cognitive activity (CRIq) | Q | Q | Q | Q | Q |
| Loneliness (de Jong Gierveld, 6-item) | Q | Q | Q | Q | Q |
| Cardiovascular diseases | Q | Q | Q | Q | Q |
| Obesity (body weight, height) | Q+PE | Q | Q | Q | Q+PE |
| Hypertension (SBP, DBP) | Q+PE | Q | Q | Q | Q+PE |
| High cholesterol (LDL, HDL, TC) | Q+FBS | Q | Q | Q | Q+FBS |
| Diabetes mellitus (glucose, HbA1C) | Q+FBS | Q | Q | Q | Q+FBS |
| Renal dysfunction (eGFR) | Q+FBS | Q | Q | Q | Q+FBS |
| Depression (CES-D) | Q | Q | Q | Q | Q |

CES-D, centre for epidemiological studies depression scale; CRIq, cognitive reserve index questionnaire (adapted); DBP, diastolic blood pressure; eGFR, estimated glomerular filtration rate; FBS, fasting blood sample; FFQ, food frequency questionnaire; HbA1C, haemoglobin A1C; HDL, high-density lipoproteins; IPAQ, international physical activity questionnaire; LDL, low-density lipoproteins; PE, physical examination; Q, online questionnaire; SBP, systolic blood pressure; SQUASH, short questionnaire to assess health-enhancing physical activity; TC, total cholesterol.

present, but the disease is managed well as participants have regular meetings with their general practitioner for disease control (diabetes mellitus) or use medication for disease management (cardiovascular diseases, hypertension, high cholesterol, renal dysfunction and depression) (see figure 1).

### Tailor-made online lifestyle advice for dementia risk reduction

Participants also receive an online tailor-made lifestyle advice targeting risk factors associated with dementia and following the Dutch guidelines for a healthy diet, alcohol consumption, physical activity, diabetes mellitus, renal dysfunction and cardiovascular health including cholesterol levels and BMI.[54–58] For each risk and protective factor, information is given about (1) the norm (cut-off point for not having this risk factor), (2) the association between the risk factor and dementia and (3) lifestyle advice how to tackle this factor. The online lifestyle advice was tailored to the participants based on (1) the presence of risk factors, (2) the strength of the association between the risk factors and dementia[19 52] and (3) the stages of change of the health behaviour-related risk factors (physical inactivity, diet, alcohol consumption, smoking behaviour, cognitive activity, social activity). The stages of change are determined by asking 'Which statement fits best for you?', where each answer option reflects one of the following stages of change: pre-contemplation, contemplation, preparation, action and maintenance.[33] It is known that participants who are in the preparation and action stage are more willing to change their health behaviour, therefore lifestyle advice for these factors are given first.[33]

In case medically relevant findings are found, including untreated diabetes mellitus (glucose ≥7.0 mmol/L or (glucose ≥6.1 mmol/L and HbA1C>53 mmol/mol)), untreated renal dysfunction (estimated glomerular filtration rate (eGFR) ≤60 mL/min/1.73 m$^2$) and increased risk for developing CVD (CVD risk ≥10% according to the Dutch Systematic Coronary Risk Evaluation (SCORE) formula[59]), participants receive, in addition to the online tailor-made lifestyle advice, a separate message in their personal inbox with the recommendation to contact their general practitioner to verify the results and discuss whether treatment is needed.

### Outcome measures and measurements

Participants are invited to fill in the online questionnaire at baseline and four times (3, 6, 9 and 12 months after baseline measurement) during 1 year follow-up. Physical examination, including the fasting blood sample for direct laboratory measurements, is only done at baseline and 12 months after baseline measurement (see online supplemental file 2). Data from the online questionnaires and physical examination are stored automatically in an electronic case report form (eCRF) data management programme, which is only accessible by the researchers involved in this study. Data from the direct laboratory measurement are entered manually in the eCRF data management programme. Every month, memory clinics are requested to provide information about (1) the number of eligible participants (eg, new cases of AD and VD), (2) the number of envelopes that are given away and (3) any difficulties with the recruitment of participants. In order to keep participating memory clinics involved

**Table 2** Definition for the three categories in the personal health profile at baseline

| Modifiable risk factors | Keep this up | Remember to manage well | Room for improvement |
|---|---|---|---|
| Diet | MIND-diet score=14 points | NA | MIND-diet score<14 points |
| Alcohol consumption | Average number of units of alcohol per week≤7 and number of units per day is: ≤3 for women or ≤4 for men | NA | Average number of units of alcohol per week>7 or number of units per day is: >3 for women or >4 for men |
| Cognitive activity | Paid working hours≥24 or CRIq score≥50 | NA | Paid working hours<24 and CRIq score<50 |
| Physical activity | (MVPA/week≥150 and sitting time≤8 hours/day) or (MVPA/week<150 and sitting time<4 hours/day) | NA | Sitting time>8 hours/day or (sitting time≥4 hours/day and MVPA/week<150) |
| Smoking | Past or never smoker | NA | Current smoker |
| Loneliness | De Jong Gierveld score<2 | NA | De Jong Gierveld score≥2 |
| CVD | No CVD | At least one CVD and receives medical treatment | At least one CVD and no medical treatment |
| Weight | BMI≥18.5 and BMI<25.0 | NA | BMI<18.5 or BMI≥25.0 |
| Blood pressure | DBP<90 mmHg and SBP<140 mmHg and no medical treatment | DBP<90 mmHg and SBP<140 mmHg and medical treatment | DBP≥90 mmHg or SBP≥140 mmHg |
| Cholesterol | LDL≤2.5 mmol/L and TC/HDL≤5 mmol/L and no medical treatment | LDL≤2.5 mmol/L and TC/HDL≤5 mmol/L and medical treatment | LDL>2.5 mmol/L or TC/HDL>5 mmol/L |
| Diabetes mellitus | Glucose<7.0 mmol/L and HbA1C≤53 mmol/mol | (HbA1C≤53 mmol/mol and medical treatment) or (glucose<7.0 mmol/L and HbA1C>53 mmol/mol and medical treatment) | (HbA1C>53 mmol/mol and no medical treatment) or (glucose≥7.0 mmol/L and HbA1C>53 mmol/mol) or (glucose≥7.0 mmol/L and HbA1C≤53 mmol/mol and no medical treatment) |
| Kidney | eGFR≥60 mL/min/1.73 m$^2$ | eGFR<60 mL/min/1.73 m$^2$ and medical treatment | eGFR<60 mL/min/1.73 m$^2$ and no medical treatment |
| Depression | CES-D<16 points | CES-D≥16 points and medical treatment | CES-D≥16 points and no medical treatment |

BMI, body mass index; CES-D, centre for epidemiological studies depression scale; CRIq, cognitive reserve index questionnaire; CVD, cardiovascular diseases; DBP, diastolic blood pressure; eGFR, estimated glomerular filtration rate; HbA1C, haemoglobin A1C; HDL, high-density lipoproteins; LDL, low-density lipoproteins; MIND-diet, mediterranean-DASH diet intervention for neurodegenerative delay; MVPA, moderate to vigorous physical activity; SBP, systolic blood pressure; TC, total cholesterol.

in the study, every 3 months newsletters are sent around and memory clinics are contacted monthly to evaluate the uptake.

## Primary outcome

The primary outcome is the difference in uptake (eg, the percentage of eligible people that signed the online informed consent form and completed a risk assessment of the total number of eligible people) between the active and passive recruitment strategy. The total number of eligible people in each recruitment group (active vs passive) are based on the number of new cases of AD or VD in all memory clinics during the recruitment period, assuming an average of one child per patient with dementia receiving the envelope with the PIF including a login access number. Due to privacy regulations, it is

not possible to collect data regarding the reasons for non-participation.

## Secondary outcomes

Secondary outcomes include:

(1) The change in LIBRA score. The LIBRA score has been validated among individuals in midlife and reflects an individual's potential to reduce their risk on developing late-onset dementia.[52] The LIBRA score consists of 12 currently known protective (ie, Mediterranean diet, low/moderate alcohol consumption, cognitive activity) and risk factors (ie, physical inactivity, smoking, CVD, hypertension, high cholesterol, diabetes mellitus, obesity, renal dysfunction, depression) for dementia and ranges from −5.9 (low risk for developing dementia) to 12.7 (high risk for developing dementia).

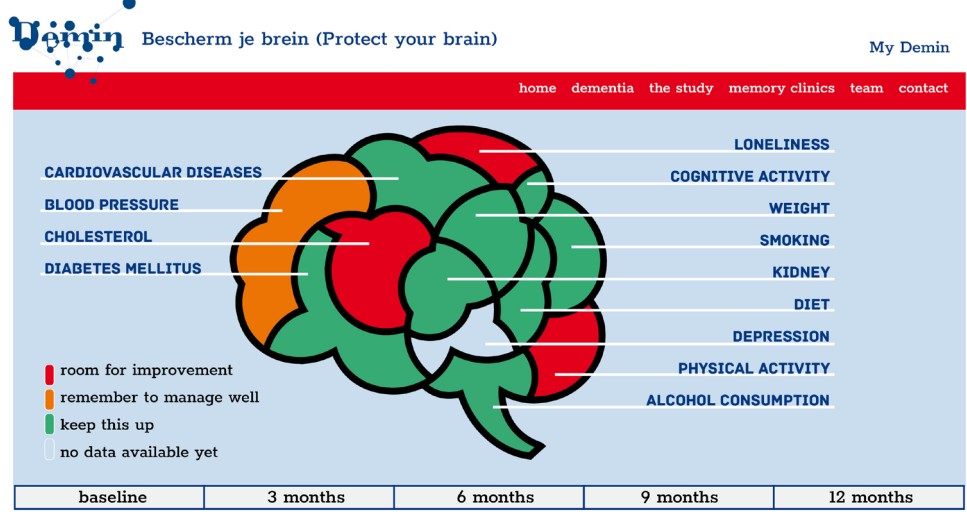

**Figure 1** An example of a personal health profile.

A one point increase in the LIBRA score is associated with a 19% higher risk for dementia.[52 60] The definitions and corresponding scores for the three protective and ten risk factors for dementia are described in table 3.

(2) The change in the individual health behaviours, including physical activity (minutes of moderate to vigorous physical activity per week), diet (mediterranean-DASH diet intervention for neurodegenerative delay score; 0–14), alcohol consumption (number of glasses of alcohol per week), smoking behaviour (current smoker (yes/no) and number of cigarettes/cigars a day), cognitive activity (leisure-time cognitive activity score and number of hours paid work), loneliness (overall loneliness score; 0–6) and social activity (number of contacts per 2 weeks) and their stage of change over time. The stages of change are categorised into pre-contemplation

| Table 3 | Definition of risk and protective factors for dementia in the LIBRA score and corresponding scores | | |
|---|---|---|---|
| **Modifiable risk factors** | | **Definition** | **Score** |
| Protective factors | | | |
| 1 | High cognitive activity | Score ≥50 points on the Cognitive Reserve Index questionnaire (leisure time activities) or hours of paid work ≥24 hours | −3.2 |
| 2 | Mediterranean diet | MIND-diet score (0–14) =14 points | −1.7 |
| 3 | Low/moderate alcohol consumption | Average number of glasses of alcohol a week ≤7 and number of glasses a day is: ≤3 glasses for women (no binge drinking), ≤4 glasses for men (no binge drinking) | −1.0 |
| Risk factors | | | |
| 4 | Cardiovascular diseases | Presence of at least one of the follow diseases: history of angina pectoris, myocardial infarction, transient ischaemic attacks, stroke or peripheral arterial diseases | +1.0 |
| 5 | Physical inactivity | Not fulfilling Dutch Norm for Physical activity defined as ≥150 min/week physical activity of moderate to vigorous intensity, measured with the SQUASH questionnaire | +1.1 |
| 6 | Renal dysfunction | Estimated glomerular filtration rate ≤60 mL/min/1.73 $m^2$ | +1.1 |
| 7 | Diabetes mellitus | Glucose (capillary blood) >7.0 mmol/L or HbA1c >53 mmol/mol | +1.3 |
| 8 | High cholesterol | LDL >2.5 mmol/L or TC/HDL >5 | +1.4 |
| 9 | Smoking | Current smoker | +1.5 |
| 10 | Obesity | BMI ≥30 | +1.6 |
| 11 | Hypertension | SBP >140 mmHg or DBP >90 mmHg | +1.6 |
| 12 | Depression | Score ≥16 points on the Centre for Epidemiologic Studies Depression scale | +2.1 |

BMI, body mass index; DBP, diastolic blood pressure; HbA1c, haemoglobin A1C; HDL, high-density lipoproteins; LDL, low-density lipoproteins; LIBRA, lifestyle for brain health; MIND-diet, mediterranean-DASH diet intervention for neurodegenerative delay; SBP, systolic blood pressure; SQUASH, short questionnaire to assess health-enhancing physical activity; TC, total cholesterol.

(1), contemplation (2), preparation (3), action (4) and maintenance (5).[33]

(3) Changes in beliefs and attitudes with regard to DRR are measured using the Motivation to Change Lifestyle and Health Behaviour for DRR Scale (MCLHB-DRR scale).[61 62] The MCLHB-DRR scale is based on the HBM,[32] which explains health-related behaviours. Seven subscales of the HBM were included: perceived susceptibility, perceived severity, perceived benefits, perceives barriers, cues to action, general health motivation and self-efficacy. Participants are asked to rate all items on a 5-point Likert scale, ranging from strongly disagree (score=1) to strongly agree (score=5). A higher score on each subscale reflects a higher motivation to change their lifestyle and health behaviour for DRR. The Dutch version of the MCLHB-DRR scale, consisting of 23 items, has shown to be valid in the Dutch general population aged between 30 and 80 years old.[63]

(4) Percentage of participants that indicated in the questionnaire that they have followed up the tailor-made online lifestyle advice ('On what risk factors did you receive lifestyle advice?' and 'Did you follow-up the tailor-made lifestyle advice since the last questionnaire (with regard to (risk factor))'?, but also the percentage of participants that indicated that they have followed up the advice to consult their General Practitioner ('Did you have contact with your general practitioner after receiving feedback on the risk and protective factors?').

### Statistical analyses

First, descriptive characteristics will be explored. The difference in uptake between the two recruitment strategies will be examined using multilevel logistic regression analyses in order to correct for clustering at memory clinic level. We will calculate the percentage with the corresponding 95% CI and use an alpha of 0.05 to test statistical significance.

The effectiveness of the online tailor-made lifestyle programme for DRR will be determined by, first comparing the change in LIBRA score, the individual risk factors and the MCLHB-DRR score between the active and passive recruitment strategy, and second comparing participants of the Demin study (active and passive recruitment strategy) to a control group consisting of Lifelines participants (large population-based cohort study (n>167 000)) (www.lifelines.nl)[64] in outcome. Lifelines participants (age 40–60 years) with a parent with dementia will be matched (using propensity scores) on non-modifiable risk factors (age, gender and education) for dementia to participants of the Demin. Subsequently, multilevel analyses will be performed to examine the change in the LIBRA score and the individual health behaviours over time. In addition, possible confounding and interaction effects will be identified and corrected for in the analysis (eg, health literacy). We will calculate relative risks with 95% CIs and use an alpha of 0.05 to test significance.

### Adverse events

The risk classification of this intervention is considered negligible, since only information and health advice is provided. Serious adverse events as a result of the intervention are not expected, thus no data safety and monitoring board is installed. Potential participants are informed about possible adverse events. For example, dementia risk assessment may help raising the awareness of their susceptibility in order to motivate health behaviour change,[32] however, risk assessment could also have an unfavourable effect. Participants may become anxious about developing dementia and could experience more stress if they receive their health profile. Therefore, participants are clearly informed that the presence or absence of risk and protective factors is not a reassurance that they will develop dementia later in life. Furthermore, participants are informed that there is the possibility that unexpected medical findings can be found. In this case, participants receive a separate message in their personal inbox with the recommendation to contact their general practitioner to verify the results (hypertension, high cholesterol, renal dysfunction, diabetes) and discuss whether treatment is needed. Participants may consider online risk assessment as a privacy risk. In this study, all personal information is kept separately from the research data, and participants use a SMS-tan code to login in their personal account.

### Patient and public involvement

Descendants of people with dementia were involved in the development of the Demin website. We assessed the knowledge, beliefs and attitudes towards dementia and dementia risk reduction among descendants of people with dementia (focus group discussions). The results of the focus group discussions were used to develop the Demin website in order to improve the participant recruitment and encourage health behaviour change among participants.

### Ethics and dissemination

This study is approved by the Dutch Ministry of Health, Welfare and Sport according to the Dutch Population Screening Act. Research which is considered to be Population Screening on the ground of the Population Screening Act, for which ministerial approval is required, does not have to be assessed on the basis of the Medical Research Involving Human Subjects Act.[65] Population screening is defined as 'medical research in persons carried out on an entire population or a category thereof aimed at the detection of certain types of disease or certain risk indicators for the benefit of the participating subjects'.[66] This project focuses on the attenuations of risk factors for dementia. Since these risk factors are merely lifestyle factors, a positive impact beyond dementia may be expected. Due to a healthy lifestyle more healthy life years are added to people's lives, which may ultimately increase the risk on dementia as age is an important risk factor for dementia. This research is conducted in accordance to the international ethical guidelines.[67]

All participants give informed consent to participate in this study, by signing an electronic informed consort form using SMS-tan (see online supplemental file 3). Authorship will be allocated using the guidelines for authorship defined by the International Committees of Medical Journal Editors.[68] The results of the trial will be submitted to an international peer-reviewed journal and presented at national and international conferences.

**Author affiliations**
[1]Department of Epidemiology, University of Groningen, University Medical Centre Groningen, Groningen, The Netherlands
[2]Department of Medical Informatics, University of Amsterdam, Amsterdam UMC, Amsterdam, The Netherlands
[3]Wenckebach Institute for Training and Education, University of Groningen, University Medical Centre Groningen, Groningen, The Netherlands
[4]Department of Neurology and Alzheimer Centre Groningen, University of Groningen, University Medical Centre Groningen, Groningen, The Netherlands
[5]Department of Health Sciences, University of Groningen, University Medical Centre Groningen, Groningen, The Netherlands
[6]Department of Psychiatry, University of Groningen, University Medical Centre Groningen, Groningen, The Netherlands
[7]Medical School Twente, Medical Spectrum Twente, Enschede, The Netherlands

**Acknowledgements** The authors would like to thank the developers of the Demin website (Bruna&Bruna, Rocket Industries, Centric and Research Data Support from the University Medical Centre Groningen) to make this research possible and the Board of Directors and the collaborators of the participating memory clinics for the local approval and collaboration to conduct this multicentre study: Albert Schweitzer Hospital (Dordrecht), Gelre Hospital (Apeldoorn), University Medical Centre Groningen (Groningen), Medical Centre Leeuwarden (Leeuwarden), Nij Smellinghe (Drachten), Isala Zwolle (Zwolle), Martini Hospital (Groningen), Haga Hospital (Den Haag), Scheper Hospital (Emmen), Refaja Hospital (Stadskanaal), St Jans Gasthuis (Weert), University Medical Centre Utrecht (Utrecht), Reinier de Graaf Gasthuis (Delft), Maxima Medical Centre (Eindhoven), Radboud University Medical Centre (Nijmegen), TweeSteden Hospital (Tilburg), Erasmus Medical Centre (Rotterdam), Ommelanden Hospital Groningen (Scheemda) and Rijnstate Hospital (Arnhem and Zevenaar).

**Collaborators** The members of the Demin consortium are: Elske Marije Abma (MD), Ameen Abu-Hanna (PhD), Erik Buskens (MD, PhD), Jürgen Claassen (MD, PhD), Peter Paul De Deyn (MD, PhD), Theo Feitsma (MD), Liesbeth Hempenius (MD, PhD), Jan Hoogmoed (MD), Ad Kamper (MD, PhD), Dineke Koek (MD, PhD), Jolijn Kragt (MD, PhD), Joep Lagro (MD), Els Lambooij (MD), Marc Langedijk (MD), Els Maeckelberghe (PhD), Francesco Mattace Raso (MD, PhD), Marieke Meinardi (MD, PhD), Richard Oude Voshaar (MD, PhD), Fransje Reesink (MD, PhD), Sophia de Rooij (MD, PhD), Antoinette Scheepmaker (MD), Nynke Smidt (PhD), Petra Spies (MD, PhD), Diana Taekema (MD, PhD), Jos Verkuyl (MD), Ralf Vingerhoets (MD), Joyce Vrijsen (MSc), Andrea de Winter (PhD).

**Contributors** JV contributed to the study concept and design, drafting of the manuscript and critical revision of the manuscript. NS and SEdR conceived the idea, were responsible for data acquisition, contributed to the study concept and design and the critical revision of the manuscript. AAH, ELMM, PPDD, AFdW, FER, RCOV and EB contributed to the study concept and design, and the critical revision of the manuscript. All authors read and approved the final manuscript.

**Funding** This study was supported by grants from the Netherlands Organisation for Health Research and Development (ZonMw), subprogram prevention program (project number: 531002008).

**Competing interests** None declared.

**Patient consent for publication** Not required.

**Provenance and peer review** Not commissioned; externally peer reviewed.

**ORCID iD**
Joyce Vrijsen http://orcid.org/0000-0003-1506-2266

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
