## [Reviewer comments · BMJ Open]

ARTICLE DETAILS

TITLE (PROVISIONAL)	The uptake and effectiveness of a tailor-made online lifestyle program targeting modifiable risk factors for dementia among middle-aged descendants of people with recently diagnosed dementia: study protocol of a cluster randomised controlled trial (Demin study)
AUTHORS	Vrijisen, Joyce; Abu-Hanna, Ameen; Maeckelberghe, Els; De Deyn, Peter Paul; de Winter, Andrea; Reesink, Fransje; Oude Voshaar, Richard; Buskens, Erik; de Rooij, Sophia; Smidt, Nynke

VERSION 1 – REVIEW

REVIEWER	Notger Müller DZNE Germany
REVIEW RETURNED	11-May-2020

GENERAL COMMENTS	I have a major concern with this study. The authors state their primary outcome is the difference in uptake between passive and active recruitment across different memory clinics. Hence, a major part of the paper should be dedicated to theories etc. related to recruitment strategies. However, there is literally no part of the paper dedicated to this point. As is, the paper reads like a study on effects of lifestyle interventions against dementia with the usual background information given on rising prevalence etc. In my opinion the paper should be completely rewritten with a focus on their primary outcome, namely recruitment strategy. Also, given recruitment is the key, I wonder whether the randomisation procedure (i.e. assigning whole memory clinics to either the passive or active arm) is really appropriate. Active recruitment crucially depends on the person that contacts eligible participants and this cannot be matched between different institutions. In my opinion, given the primary outcome, randomisation should be performed within a institution.
--

REVIEWER	Gloria Wong The University of Hong Kong, Hong Kong
REVIEW RETURNED	19-May-2020

GENERAL COMMENTS	This protocol paper describes a cluster randomised controlled trial on the effectiveness of programme uptake in active versus passive methods in recruiting middle-aged people with familiar risk of dementia (having one of the parents recently diagnosed as having Alzheimer's or vascular dementia) into an online tailored lifestyle programme. The study will also investigate – as secondary outcomes – the effectiveness of the programme in reducing lifestyle risk (both composite score and individual risk factors) by comparing this population with a matched control group from another cohort.
--

	The paper is clearly written and follows the standards of reporting clinical trial protocols. Only a few minor points need to be clarified: 1. It would be helpful if the authors could explain the reasons for using 'uptake' as the only primary outcome, and 'effectiveness in risk reduction' as secondary outcomes. It would appear that the secondary research questions would have much greater potential impact in terms of both scientific innovations and clinical implications, and if the effectiveness of the risk reduction programme is not yet known, it is uncertain how important it is to understand the uptake of the programme. 2. In relation to 1, currently the authors touched on this point briefly in the introduction (p.8, line 133, "A challenge of health behaviour change..."), but the discussion is about uptake of screening programmes instead of that for healthy lifestyle. The two (screening and lifestyle change) are quite different in concept and in practice (e.g., there are discussions about the ethics of screening for dementia or dementia risk, including the predictive values of such screening tests and the available treatment; such ethical considerations may not always be relevant to healthy lifestyle such as promotion of smoking cessation, which is relevant for universal prevention). The authors may explain how this current programme map onto these concepts of prevention, so that the readers can better appreciate the importance of the primary outcome of 'uptake'. 3. As the authors have pointed out on page 8, second paragraph, the current empirical evidence about the effectiveness of targeting modifiable factors in preventing or delaying dementia onset is still thin, despite growing evidence of the association between modifiable factors and dementia. This lack of evidence is possibly due to the challenges in research design that can delineate causal relationship and intervention effects (with midlife risk reduction programmes requiring decades of follow-up of a large sample). As such, whether risk reduction programmes such as the current one can effectively prevent or delay the onset of dementia remains an important and unanswered question, which is not possible to address in this study. It would be helpful if the authors could comment on how this study could contribute to this knowledge indirectly, and whether or not (and if not, the reasons why not, e.g. short follow-up period) they will analyse the incidence of MCI or dementia onset in their study. 4. For the authors' consideration, given the current focus on modifiable lifestyle factors, a 2019 paper could be relevant: based on data from the Swedish National Study on Aging and Care in Kungsholmen (SNAC-K), engaging in cognitive reserve-enhancing activities appears to mitigate the genetic risk of dementia attributable to APOE-ε4. This provide further support to a focus in lifestyle changes. Reference: Dekhtyar S, Marseglia A, Xu W, Darin-Mattsson A, Wang HX, Fratiglioni L. Genetic risk of dementia mitigated by cognitive reserve: A cohort study. Ann Neurol. 2019;86(1):68-78. 5. As the programme emphasizes tailored advice, the authors may comment on the current availability of evidence for tailoring, by individual characteristics such as gender and pre-existing lifestyle pattern and preference. Given that many healthy lifestyle behaviours are linked with a wide range of individual characteristics that are also related to dementia risk (e.g., education level, health literacy and engagement in cognitively stimulating activities), there is a possibility that different domains of risk reduction can impact on a person's
--	---

	dementia risk differently. Nevertheless, it is unlikely that the current evidence could support this level of tailoring, although this could be a point to note for programmes that aspire to be tailor-made. In relation to this, the authors could also mention whether and how emerging evidence regarding lifestyle risks would be handled in this programme, in terms of advice to participants as well as risk score calculation. 6. The authors could comment on whether the study design for the 'uptake' question may (or may not) introduce any bias that undermines the study's validity in answering 'effectiveness of risk reduction' question. In particular, would the authors comment on the likelihood that participants from the active and the passive recruitment groups differ in baseline characteristics such as health literacy and motivation (e.g., those who would be willing to complete a lengthy assessment upon seeing a leaflet might be more health conscious than those being talked into the programme?), which could affect their adherence to and thus the effectiveness of the risk reduction advice provided in the online programme. In the propensity score matching with Lifelines participants, the authors suggested using only non-modifiable risk factors. What are the considerations for not including baseline health literacy in the matching?
--	--

VERSION 1 – AUTHOR RESPONSE

Response to the Reviewers' comments

Reviewer 1

Point 1: (BACKGROUND) I have a major concern with this study. The authors state their primary outcome is the difference in uptake between passive and active recruitment across different memory clinics. Hence, a major part of the paper should be dedicated to theories etc. related to recruitment strategies. However, there is literally no part of the paper dedicated to this point. As is, the paper reads like a study on effects of lifestyle interventions against dementia with the usual background information given on rising prevalence etc. In my opinion the paper should be completely rewritten with a focus on their primary outcome, namely recruitment strategy.

Response 1: We are sorry to hear that the reviewer has a major concern with the presentation of our study. Perhaps, our study design is not entirely clear. Based on theories on recruitment strategies [4], first the number of potential eligible participants of each memory clinics was registered over a 2 month period and the options how to recruit potential participants in each memory clinic were discussed (for active and passive recruitment) with the staff of each memory clinic (for example, who is giving the participant information form and who is checking the eligibility of potential participants (i.e. having a parent with AD or VD). The aim was to assess whether indeed thus far healthy individuals confronted with an alarming diagnosis among their parents, would be willing to consider implications for them personally. Accordingly, it remains unclear whether it would be feasible to recruit potential participants at the memory clinic, if the dementia risk reduction program is effective in tackling the risk factors for dementia. Therefore, we focussed on the feasibility of such a program. Is an active recruitment strategy necessary or is a passive recruitment strategy sufficient? In our opinion, information regarding our primary outcome, which is uptake, was already included in the introduction but has been extended now.

We have added the following sentences to the Background section:

- (See page 6, Lines 148-157): The two main types of strategies for recruitment are the active and passive recruitment strategy. Active recruitment involves a personal

invitation by the project staff and healthcare providers (e.g. proactive) and passive recruitment involves recruitment of participants through various channels such as flyers and advertisements (e.g. reactive) [4]. The most effective recruitment strategy is proactive referral from a healthcare provider, while displaying posters and flyers showed to be less effective [5].

- (See page 7, Lines 163-170): A previous study showed that the majority of the Dutch general population is unaware of the relationship between modifiable risk factors and brain health, particularly regarding major cardiovascular risk factors (e.g. hypertension, hypercholesterolemia and coronary heart disease) [6]. Having a parent who is recently diagnosed with AD or VD could have led to an increased knowledge on dementia and risk perception [7]. Therefore, middle-aged descendants of recently diagnosed people with AD or VD might be receptive to assess their risk and motivated to adopt a healthier lifestyle as they just realized their (familial) risk [7,8].

- (See page 7, Lines 176-179): This will give insight in to what extent it is feasible to recruit middle-aged descendants of people with AD or VD at the memory clinic and whether these potential participants are willing to participate in a tailor-made online lifestyle program in order to reduce their dementia risk.

Point 2: (METHOD) Also, given recruitment is the key, I wonder whether the randomisation procedure (i.e. assigning whole memory clinics to either the passive or active arm) is really appropriate. Active recruitment crucially depends on the person that contacts eligible
4

participants and this cannot be matched between different institutions. In my opinion, given the primary outcome, randomisation should be performed within an institution.

Response 2: We decided to randomize the memory clinics and not the participants to prevent contamination between the two interventions (active and passive recruitment strategy), as is recommended [9]. Interested memory clinics could only participate in this study when they did not have a preference for a particular recruitment strategy in order to prevent a difference in motivation to recruit eligible participants. Additionally, the allocated recruitment strategy was not announced to the memory clinic after confirmation of participation was received and local approval was obtained.

Further, since we are interested whether it would be feasible to recruit potential participants at the memory clinic, it is important to investigate uptake in real-life conditions where active recruitment can differ between persons (i.e. staff) that contact eligible participants. In addition to the recruitment rate (e.g. uptake), we will also collect information about the number of diagnoses with AD and/or VD and the number of eligible participants (e.g. patients without children do not provide an eligible participant) every month, and whether eligible participants received the information letter. This will give insight in the performance in recruitment of each memory clinic.

Reviewer 2

Point 1: (BACKGROUND) It would be helpful if the authors could explain the reasons for using 'uptake' as the only primary outcome, and 'effectiveness in risk reduction' as secondary outcomes. It would appear that the secondary research questions would have much greater potential impact in terms of both scientific innovations and clinical implications, and if the effectiveness of the risk reduction programme is not yet known, it is uncertain how important it is to understand the uptake of the programme.

Response 1: We thank the reviewer for this comment and have clarified the added value of investigating the uptake of a dementia risk reduction program among middle-aged descendants of people with AD or VD. The aim was to assess whether indeed thus far healthy individuals confronted with an alarming diagnosis among their parents, would be willing to consider implications for them personally. Accordingly, it remains unclear whether it would be feasible to recruit potential participants at the memory clinic, if the dementia risk

reduction program is effective in tackling the risk factors for dementia. Therefore, we focussed on the feasibility of implementing such a program. Is an active recruitment strategy necessary or is a passive recruitment strategy sufficient? (see also our reply to reviewer 1, point 1)

We have added the following sentences to the Background section:

- (see page 7, Lines 163-170): A previous study showed that the majority of the Dutch general population is unaware of the relationship between modifiable risk factors and brain health, particularly regarding major cardiovascular risk factors (e.g. hypertension, hypercholesterolemia and coronary heart disease) [35]. Having a parent who is recently diagnosed with AD or VD could have led to an increased knowledge on dementia and risk perception [38]. Therefore, middle-aged descendants of recently diagnosed people with AD or VD might be receptive to assess their risk and motivated to adopt a healthier lifestyle as they just realized their (familial) risk [7,8].

- (See page 7, Lines 176-179): This will give insight in to what extent it is feasible to recruit middle-aged descendants of people with AD or VD at the memory clinic and whether these potential participants are willing to participate in a tailor-made online lifestyle program in order to reduce their dementia risk.

Point 2: (BACKGROUND) In relation to 1, currently the authors touched on this point briefly in the introduction (p.8, line 133, "A challenge of health behaviour change..."), but the discussion is about uptake of screening programmes instead of that for healthy lifestyle. The two (screening and lifestyle change) are quite different in concept and in practice (e.g., there are discussions about the ethics of screening for dementia or dementia risk, including the predictive values of such screening tests and the available treatment; such ethical considerations may not always be relevant to healthy lifestyle such as promotion of smoking cessation, which is relevant for universal prevention). The authors may explain how this current programme map onto these concepts of prevention, so that the readers can better appreciate the importance of the primary outcome of 'uptake'.

Response 2: We thank the reviewer for this comment and added information on the ethical considerations for dementia risk assessment (e.g. assessing risk and protective factors) in the background section, so that the readers can better appreciate the importance of the primary outcome as mentioned above (see Response 1). In the Adverse events section (see page 21, Lines 446-458), we provided information about how the current study assures that potential participants make a well-informed decision whether to assess their risk and protective factors and participate in the program.

6

We have added the following sentences to the Background section (see page 7, Lines 160-163):

These factors are essential and useful to make a well-informed decision about dementia risk assessment. Therefore, information on dementia, the risk and protective factors for dementia, heritability, and how to tackle risk and protective factors for dementia are important factors in the development of a web-based intervention.

Point 3: (BACKGROUND) As the authors have pointed out on page 8, second paragraph, the current empirical evidence about the effectiveness of targeting modifiable factors in preventing or delaying dementia onset is still thin, despite growing evidence of the association between modifiable factors and dementia. This lack of evidence is possibly due to the challenges in research design that can delineate causal relationship and intervention effects (with midlife risk reduction programmes requiring decades of follow-up of a large sample). As such, whether risk reduction programmes such as the current one can effectively prevent or delay the onset of dementia remains an important and unanswered question, which is not possible to address in this study. It would be helpful if the authors could comment on how this study could contribute to this knowledge indirectly, and whether

or not (and if not, the reasons why not, e.g. short follow-up period) they will analyse the incidence of MCI or dementia onset in their study.

Response 3: The reduction of the dementia incidence is a future goal, however in order to determine the effect of the dementia risk reduction, an online lifestyle program on the incidence would indeed take a long follow up of approximately 20 years or even more. We agree with the reviewer that a baseline measure of cognition would be very valuable. Initially, we planned to include two cognition tests in the dementia risk assessment. However, the Dutch Health Council (the organ which advises the Ministry of Health, Sports and Welfare for approval) advised against including cognitive tests, as this could give the participants the impression that we screen on dementia or cognitive decline

(<https://www.healthcouncil.nl/latest/news/2018/06/12/population-screening-act-study-of-online-lifestyle-advice-for-reducing-the-risk-of-dementia>).

They believed that this might cause feelings of anxiety, as currently there is still no treatment to cure dementia. Nevertheless, we do know that approximately 35% of dementia cases can be attributed to several modifiable risk factors that were included in this study [10] (see page 5, Lines 109-116). This shows great potential for dementia prevention, and measuring potential effectiveness on a shorter notice. The findings of this study could help make policy makers set their priorities for dementia risk reduction and healthy ageing in general.

Primarily, the findings of this study will show whether middle-aged descendants of people with AD or VD are sensitive for adaption of health risk behaviour. This will affect dementia risk, and also avert the development of other chronic diseases, such as cardiovascular diseases.

We have added the following sentences to the Background section (see page 6, Lines 134-136):

However, since dementia is mainly prevalent among older people, a long follow-up period of approximately 20 years would be required in order to determine the effectiveness of lifestyle interventions on preventing or postponing the onset of dementia and reducing the age-related dementia incidence [11–13].

Point 4: For the authors' consideration, given the current focus on modifiable lifestyle factors, a 2019 paper could be relevant: based on data from the Swedish National Study on Aging and Care in Kungsholmen (SNAC-K), engaging in cognitive reserve-enhancing activities appears to mitigate the genetic risk of dementia attributable to APOE-ε4. This provide further support to a focus in lifestyle changes.

7

Reference: Dekhtyar S, Marseglia A, Xu W, Darin-Mattsson A, Wang HX, Fratiglioni L. Genetic risk of dementia mitigated by cognitive reserve: A cohort study. *Ann Neurol*. 2019;86(1):68-78.

Response 4: We thank the reviewer for sharing this interesting paper on the interaction effects between the APOE e4 allele and modifiable risk factors. Indeed, genotyping might have added to a more precise estimation of risk. However, as alluded above, the Dutch Health Council was afraid that this program would give the impression that we screen on dementia, which can cause feelings of anxiety since there is no treatment, so we refrained from including it. Also, APOE genotyping would have considerable costs attached to it. Possibly, tailoring the online lifestyle advice on APOE genotype would be an interesting addition for future research.

Point 5: As the programme emphasizes tailored advice, the authors may comment on the current availability of evidence for tailoring, by individual characteristics such as gender and pre-existing lifestyle pattern and preference. Given that many healthy lifestyle behaviours are linked with a wide range of individual characteristics that are also related to dementia risk (e.g., education level, health literacy and engagement in cognitively stimulating activities), there is a possibility that different domains of risk reduction can impact on a person's dementia risk differently. Nevertheless, it is unlikely that the current evidence could support

this level of tailoring, although this could be a point to note for programmes that aspire to be tailor-made. In relation to this, the authors could also mention whether and how emerging evidence regarding lifestyle risks would be handled in this programme, in terms of advice to participants as well as risk score calculation.

Response 5: Web-based tailored interventions are increasingly common in promoting a healthy lifestyle. However, there is a considerable variation in the meaning of 'tailored'[14]. In this study, tailoring an health advice to several characteristics of individual study participants (e.g. presence of risk factors and the stages of change) was one of the employed techniques to optimize the effectiveness of the program for health behavior change [15,16] (see page 17, Lines 329-332). It is known that participants who are in the preparation and action stage (measured by the stages of changes) are more willing to change their health behaviour, therefore lifestyle advice for these factors are given first [17]. We choose not to communicate the weight of the risk factors (according to the LIBRA score) to the participants, since this can be very demotivating. For example, if participants improve on two risk factors (-2.0) but their cognitive activity got worse (+3.2), then the LIBRA score will be higher, indicating a higher risk for dementia.

Furthermore, we did not tailor the health advice to their level of health literacy, but we do provide all participants with both textual information as audiovisual information (e.g. spoken animation), which is the best way to communicate complex health information to people with low health literacy. As animations do not negatively influence high literate audiences, information adapted to audiences with low health literacy suits people with high health literacy as well [18].

Point 6: The authors could comment on whether the study design for the 'uptake' question may (or may not) introduce any bias that undermines the study's validity in answering 'effectiveness of risk reduction' question. In particular, would the authors comment on the likelihood that participants from the active and the passive recruitment groups differ in baseline characteristics such as health literacy and motivation (e.g., those who would be willing to complete a lengthy assessment upon seeing a leaflet might be more health conscious than those being talked into the programme?), which could affect their adherence to and thus the effectiveness of the risk reduction advice provided in the online programme. In the propensity score matching with Lifelines participants, the authors suggested using only 8

non-modifiable risk factors. What are the considerations for not including baseline health literacy in the matching?

Response 6: We agree with the reviewer that health literacy is indeed an important issue for both uptake and effectiveness and therefore included the health literacy measure 'STOFHLA' in the baseline questionnaire [19,20]. Regarding the uptake, to enhance comparability between the intervention (participants of the active recruitment strategy) and control group (participants of the passive recruitment strategy), the memory clinics were matched into pairs based on the following criteria: (i) number of newly diagnosed dementia (VD, AD or mixed dementia) patients seen per year (range vary from 60 to 350 patients per year) and (ii) the average social economic position (SEP) of the population living around the memory clinic (neighbourhood SEP), based on data from Statistics Netherlands [21].

Subsequently, the matched memory clinics were randomized (pairwise randomization) to the active of passive recruitment strategy. In addition, as mentioned above health information is provided by written text and in an audio-visual format such as a spoken animation, to assure inclusion or participants with different levels of health literacy [18].

Regarding the effectiveness, we are aware that the two recruitment strategies might lead to a difference in patient characteristics. Therefore, we will also compare the effectiveness of the online tailor-made lifestyle program for dementia risk reduction by comparing the change in LIBRA score, the individual risk factors and the MCLHB-DRR score between the active and passive recruitment strategy.

We did not consider including health literacy in the matching process since health literacy and educational level are often highly correlated. Our expectation is that any disbalance between baseline characteristics due to a difference in health literacy or educational level, will probably diminish by matching on educational level. Nevertheless, we will include health literacy as covariate in the analyses. First, we will check whether the study participants differ in baseline characteristics (SMD > 0.2) (e.g. age, gender, education and health literacy) with the participants of the control group (Lifelines participants) and whether matching additionally on health literacy will improve the balance between the two groups. Further, we will explore to what extent health literacy influences health behavior change.

REFERENCES

- 1 Central Committee on Research Involving Human Subjects. Population Screening Act. 2020. <https://english.ccmo.nl/investigators/legal-framework-for-medical-scientificresearch/laws/population-screening-act> (accessed 18 Jun 2020).
- 2 Central Committee on Research Involving Human Subjects. The definition of population screening. 2020. <https://english.ccmo.nl/investigators/types-of-research/other-types-of-research/population-research/definition> (accessed 24 Jun 2020).
- 3 Council for International Organizations of Medical Sciences. International Ethical Guidelines for Biomedical Research Involving Human Subjects. *Bull Med Ethics* 2002;182:17–23.
- 4 Yancey AK, Ortega AN, Kumanyika SK. EFFECTIVE RECRUITMENT AND RETENTION OF MINORITY RESEARCH PARTICIPANTS. *Annu Rev Public Health* 2006;27:1–28. doi:10.1146/annurev.publhealth.27.021405.102113
- 5 Bracken K, Askie L, Keech AC, et al. Recruitment strategies in randomised controlled trials of men aged 50 years and older: a systematic review. *BMJ Open* 2019;9. doi:10.1136/BMJOPEN-2018-025580
- 6 Heger I, Deckers K, van Boxtel M, et al. Dementia awareness and risk perception in middle-aged and older individuals: baseline results of the MijBreincoach survey on the association between lifestyle and brain health. *BMC Public Health* 2019;19:678. doi:10.1186/s12889-019-7010-z
- 7 Andersson L, Stanich J. Life events and their impact on health attitudes and health behavior. *Arch Gerontol Geriatr* 1996;23:163–77. doi:10.1016/0167-4943(96)00716-9
- 8 Rosenberg A, Coley N, Soulier A, et al. Experiences of dementia and attitude towards prevention: a qualitative study among older adults participating in a prevention trial. *BMC Geriatr* 2020;20:99. doi:10.1186/s12877-020-1493-4
- 9 Schellings R, Kessels AG, ter Riet G, et al. Indications and requirements for the use of prerandomization. *J Clin Epidemiol* 2009;62:393–9. doi:10.1016/j.jclinepi.2008.07.010
- 10 Livingston G, Sommerlad A, Orgeta V, et al. The Lancet Commissions Dementia prevention, intervention, and care. *Lancet* 2017;390:2673–734. doi:10.1016/S0140-6736(17)31363-6
- 11 Singh-Manoux A, Kivimaki M, Glymour MM, et al. Timing of onset of cognitive decline: results from Whitehall II prospective cohort study. *BMJ* 2012;344:d7622. doi:10.1136/BMJ.D7622
- 12 Sperling RA, Aisen PS, Beckett LA, et al. Toward defining the preclinical stages of Alzheimer's disease: recommendations from the National Institute on Aging-Alzheimer's Association workgroups on diagnostic guidelines for Alzheimer's disease. *Alzheimers Dement* 2011;7:280–92. doi:10.1016/j.jalz.2011.03.003
- 13 Rajan KB, Wilson RS, Weuve J, et al. Cognitive impairment 18 years before clinical diagnosis of Alzheimer disease dementia. *Neurology* 2015;85:898–904. doi:10.1212/WNL.0000000000001774
- 14 Kreuter MW, Skinner CS. Tailoring: what's in a name? *Health Educ Res* 2000;15:1–4.

doi:10.1093/her/15.1.1

15 Webb TL, Joseph J, Yardley L, et al. Using the internet to promote health behavior change: a systematic review and meta-analysis of the impact of theoretical basis, use of behavior change techniques, and mode of delivery on efficacy. *J Med Internet Res* 2010;12:e4. doi:10.2196/jmir.1376

16 Krebs P, Prochaska JO, Rossi JS. A meta-analysis of computer-tailored interventions for health behavior change. *Prev Med (Baltim)* 2010;51:214–21. doi:10.1016/j.ypmed.2010.06.004

17 Prochaska JO, Velicer WF. The transtheoretical model of health behavior change. *Am J Health Promot* 1997;12:38–48. <http://www.ncbi.nlm.nih.gov/pubmed/10170434> (accessed 24 Feb 2017).

18 Meppelink CS, Weert JC van, Haven CJ, et al. The Effectiveness of Health Animations 10

in Audiences With Different Health Literacy Levels: An Experimental Study. *J Med Internet Res* 2015;17. doi:10.2196/jmir.3979

19 Baker DW, Williams M V, Parker RM, et al. Development of a brief test to measure functional health literacy. *Patient Educ Couns* 1999;38:33–42. doi:10.1016/S0738-3991(98)00116-5

20 Twickler M, Hoogstraaten E, Reuwer AQ, et al. Laaggeletterdheid en beperkte gezondheidsvaardigheden vragen om een antwoord in de zorg. *Ned Tijdschr Geneesk* 2009;153. <https://www.ntvg.nl/system/files/publications/a250.pdf> (accessed 18 Jun 2020).

21 Centraal Bureau voor de Statistiek. Percentage hoger opgeleiden van 15 jaar en ouder - Bevolkingskern (2011).

2011. http://www.cbsinuwbuurt.nl/#bevolkingskern2011_percentage_inwoners_hogere_opleiding (accessed 18 Feb 2019).

VERSION 2 – REVIEW

REVIEWER	Notger Müller DZNE Germany
REVIEW RETURNED	09-Jul-2020

GENERAL COMMENTS	Although I voted "accept" I am still puzzled by the selection of the primary outcome and how it is assessed. Whilst I agree that "uptake" is a very important issue in prevention research (the best intervention does not work if no-one participates in it) I think the authors' choice to simply compare a passive and active recruiting does not add much to our knowledge here as the result is already clear: active recruitment will be more efficient (which is already reflected in the number of clinics chosen for active vs. passive recruitment). It would have been much more interesting to compare different ways of either active or passive recruiting. Also information from eligible participants as to why they chose to NOT participate would have been crucial for assessing the effectiveness of a recruitment strategy. With that said the other merits of the study nevertheless call for its publication.
--

REVIEWER	Gloria Wong The University of Hong Kong, Hong Kong
REVIEW RETURNED	18-Jul-2020

GENERAL COMMENTS	-The authors have addressed my comments partially, in particular, as also pointed out by reviewer 1, the current presentation of the
--

	theoretical background related to the primary object is very brief, and the authors added little information directly justifying the need for this research question to be asked. In my opinion, the theoretical background, current evidence, and knowledge gap related to the effectiveness of active vs passive strategies in promoting uptake of healthy lifestyle still needs to be clearly discussed.  - It is possible that the comments were not clear to the authors, resulting in responses that are partial and indirect. For example, in Response 4, it seems that the authors have mistaken the comment as a suggestion to include genotyping, when it was meant to remind the authors to include the mentioning of an important evidence in support of the need of this study. Please add this to the background. - For Responses 4 to 6, please indicate where the clarifications have been added to the manuscript.
--	--

VERSION 2 – AUTHOR RESPONSE

Reviewer 1

Point 1: Although I voted "accept" I am still puzzled by the selection of the primary outcome and how it is assessed. Whilst I agree that "uptake" is a very important issue in prevention research (the best intervention does not work if no-one participates in it) I think the authors' choice to simply compare a passive and active recruiting does not add much to our knowledge here as the result is already clear: active recruitment will be more efficient (which is already reflected in the number of clinics chosen for active vs. passive recruitment). It would have been much more interesting to compare different ways of either active or passive recruiting. Also information from eligible participants as to why they chose to NOT participate would have been crucial for assessing the effectiveness of a recruitment strategy. With that said the other merits of the study nevertheless call for its publication.

Response 1: We regret that our reply was not sufficient to the reviewer. We have now provided more information to clarify the added value of comparing an active and passive recruitment strategy. There are multiple reasons why we are interested in comparing an active and passive recruitment strategy in Dutch memory clinics. First, we do this in order to see whether the inclusion of middle-aged descendants of recently diagnosed people with AD or VD, significantly differs between an active and passive recruitment strategy (e.g. uptake). Although previous studies have shown that an active recruitment strategy might be more effective in recruiting individuals, individuals recruited by the passive recruitment strategy might be more intrinsically motivated to participate in the study. Furthermore, individuals who

are intrinsically motivated to participate in the tailor-made online lifestyle program for dementia risk reduction might also be more likely to adopt and maintain a healthy behaviour and less likely to drop out from the study. Ultimately, the results should help us balance the advantages and disadvantages of an active recruitment strategy (higher uptake, but higher workload of staff members and probably less motivated group of individuals to adopt and maintain a healthy lifestyle).

Unfortunately, due to privacy regulations it is not possible to investigate the reasons for nonparticipation

in both groups, active and passive recruitment strategy. However, staff members of memory clinics using an active recruitment strategy have more insight in why individuals do not participate (e.g. not having children in the age of 40 to 60 years, do not understand the Dutch language). From the literature, we know that lack of time, entrenched attitudes and behaviours, restrictions in the physical environment, low socioeconomic status and lack of knowledge are barriers for potential participants to adopt a healthy lifestyle (Kelly et al. 2016).

We have added the following sentence to the Background and Method section

- (see page 7, Lines 153-155): It is shown that this lack of knowledge is a barrier to the uptake and maintenance of healthy behaviours for middle-aged individuals [43].

- (see page 7, Lines 158-161): Although we expect that the uptake in the active recruitment strategy will be higher compared to the passive recruitment strategy, participants recruited via the passive recruitment strategy might be more intrinsically motivated to adopt and maintain their healthy lifestyle and less likely to drop out of the study.

- (see page 18, Lines 360-361): Due to privacy regulations it is not possible to collect data regarding the reasons for non-participation.

Reviewer 2

Point 1: The authors have addressed my comments partially, in particular, as also pointed out by reviewer 1, the current presentation of the theoretical background related to the primary object is very brief, and the authors added little information directly justifying the need for this research question to be asked. In my opinion, the theoretical background,

current evidence, and knowledge gap related to the effectiveness of active vs passive strategies in promoting uptake of healthy lifestyle still needs to be clearly discussed.

Response 1: We regret that our reply was not sufficient to the reviewer. We have now provided more information to clarify the added value of comparing an active and passive recruitment strategy (e.g. knowledge gap) and to justify the need for this research question to be asked. See Response 1 for Reviewer 1 for an overview of the reasons why we are interested in comparing an active and passive recruitment strategy.

We have added the following sentence to the Background section

- (see page 7, Lines 153-155): It is shown that this lack of knowledge is a barrier to the uptake and maintenance of healthy behaviours for middle-aged individuals [43].

- (see page 7, Lines 158-161): Although we expect that the uptake in the active recruitment strategy will be higher compared to the passive recruitment strategy, participants recruited via the passive recruitment strategy might be more intrinsically motivated to adopt and maintain their healthy lifestyle and less likely to drop out of the study.

Point 2: It is possible that the comments were not clear to the authors, resulting in responses that are partial and indirect. For example, in Response 4, it seems that the authors have mistaken the comment as a suggestion to include genotyping, when it was meant to remind the authors to include the mentioning of an important evidence in support of the need of this study. Please add this to the background.

Response 2: We thank the reviewer for this comment and have added this evidence to the background section.

We have added the following sentence to the Background section

- (see page 5, Line 107): Therefore, a healthy lifestyle might be beneficial for individuals with a positive family history, especially for APOE ϵ 4 carriers [15–18].

Point 3: For Responses 4 to 6, please indicate where the clarifications have been added to the manuscript.

Response 3: Regarding previous Point 4, we thank the reviewer for sharing this interesting paper on the interaction effects between the APOE ϵ 4 allele and modifiable risk factors. We have added this evidence to the background section (see Response 2). We will discuss this point in more detail in the discussion section of the manuscript including the results of the

current study.

Regarding previous Point 5, tailoring the health advice to several characteristics of individual study participants (e.g. presence of risk factors and the stages of change) was one of the employed techniques to optimize the effectiveness of the program for health behavior change [15,16]. It is known that participants who are in the preparation and action stage (measured by the stages of changes) are more willing to change their health behaviour, therefore lifestyle advice for these factors are given first [17]. This information was already included in the Method section (see page 17, Lines 322-330).

Further, we choose to give an overview of the presence of the risk and protective factors for dementia and not to communicate the weight of the risk factors (according to the LIBRA score) to the participants, since this can be very demotivating. This information was already included in the Method section, but we have added a sentence for clarification (see page 13, Lines 297-298).

Furthermore, we did not tailor the health advice to their level of health literacy, but we do provide all participants with both textual information as audiovisual information (e.g. spoken animation), which is the best way to communicate complex health information to people with low health literacy. As animations do not negatively influence high literate audiences, information adapted to audiences with low health literacy suits people with high health literacy as well [18]. This information was already included in the Method section (see page 10, Lines 248-257).

Regarding previous Point 6, we agree with the reviewer that health literacy is indeed an important issue for both uptake and effectiveness and therefore included the health literacy measure 'S-TOFHLA' in the baseline questionnaire [18,19]. Regarding the uptake, to enhance comparability between the intervention (participants of the active recruitment strategy) and control group (participants of the passive recruitment strategy), the memory clinics were matched into pairs. Information on the randomization of the memory clinics was already included in the Method section (see page 9, Lines 207-219).

Regarding the effectiveness, we are aware that the two recruitment strategies might lead to a difference in patient characteristics. Therefore, we will also compare the effectiveness of the

online tailor-made lifestyle program for dementia risk reduction by comparing the change in LIBRA score, the individual risk factors and the MCLHB-DRR score between the active and passive recruitment strategy. This information was also already included in the Method section (see page 21, Lines 412-414).

We did not consider including health literacy in the matching process since health literacy and educational level are often highly correlated. Our expectation is that any disbalance between baseline characteristics due to a difference in health literacy or educational level, will probably diminish by matching on educational level. Nevertheless, we will include health literacy as covariate in the analyses if identified as confounding variable. We have added health literacy as an example of other potential confounders that we will identify and correct for.

We have added the following sentences to the Method section

- (see page 13, Lines 296-298): The personal health profile gives an overview of the presence of the risk and protective factors for dementia, without including the weight of the risk and protective factors.

- (see page 21, Line 420-421): In addition, possible confounding and interaction effects will be identified and corrected for in the analysis (e.g. health literacy).